# An archaeomagnetic study of the Ishtar Gate, Babylon

**Anita Di Chiara[1], Lisa Tauxe[2]\*, Helen Gries[3], Barbara Helwing[3], Matthew D. Howland[2,4], E. Ben-Yosef[2]**

**1** Istituto Nazionale di Geofisica e Vulcanologia, Rome (IT), Roma, Italy, **2** Scripps Institution of Oceanography, La Jolla, CA, United States of America, **3** Staatliche Museen zu Berlin, Berlin, Germany, **4** Wichita State University, Wichita, KS, United States of America

\* ltauxe@ucsd.edu

**Data Availability Statement:** Data are available at https://earthref.org/MagIC/19876/8b970b5b-39bd-4e39-81c6-6e8047e6b20a.

**Funding:** This work was supported in part by US-Israelli Binational Science Foundation Grant (bsf.

## Abstract

Data from the marriage of paleomagnetism and archaeology (archaeomagnetism) are the backbone of attempts to create geomagnetic field models for ancient times. Paleointensity experimental design has been the focus of intensive efforts and the requirements and short-comings are increasingly well understood. Some archaeological materials have excellent age control from inscriptions, which can be tied to a given decade or even a specific year in some cases. In this study, we analyzed fired mud bricks used for the construction of the Ishtar Gate, the entrance complex to the ancient city of Babylon in Southern Mesopotamia. We were able to extract reliable intensity data from all three phases of the gate, the earliest of which includes bricks inscribed with the name of King Nebuchadnezzar II (605 to 562 BCE). These results (1) add high quality intensity data to a region relatively unexplored so far (Southern Mesopotamia), (2) contribute to a better understanding of paleosecular variation in this region, and the development of an archaeomagnetic dating reference for one of the key regions in the history of human civilizations; (3) demonstrate the potential of inscribed bricks (glazed and unglazed), a common material in ancient Mesopotamia, to archaeomagnetic studies; and (4) suggest that the gate complex was constructed some time after the Babylonian conquest of Jerusalem, and that there were no substantial chronological gaps in the construction of each consecutive phase. The best fit of our data (averaging 136±2.1 ZAm²) with those of the reference curve (the Levantine Archaeomagnetic Curve) is 569 BCE.

## Introduction

Paleomagnetism and archaeology have worked together since the seminal investigations of Thellier [1] following the original suggestion by Folgheraiter [2]. The combination, known as archaeomagnetism, has benefited both fields significantly. Archaeomagnetic data provided critical constraints for the construction of geomagnetic field models which now stretch back 100 kyr (e.g., [3]) and they have helped inform discussions of thorny dating problems in archaeology (e.g., [4]). Yet, despite decades of intensive efforts, particularly in Europe and the Middle East, there are substantial open issues regarding the reliability of both paleomagnetic

org.il) 2018305 to LT and EBY. The funders had no role in study design, data collection and analysis, decision to publish, or preparation of the manuscript.

**Competing interests:** The authors have declared that no competing interests exist.

vector data (particularly the intensity) and their age constraints. While enormous effort has been put in to improving the paleointensity experiment itself (see, e.g., [5]) and understanding the sources of uncertainty in the experiment (e.g., [6]), understanding and improving the uncertainties in the age constraints for the archaeological materials remain a stubborn problem (e.g., [7, 8]).

Dates for archaeological materials are typically based on radiocarbon samples in close association with the archaeomagnetic materials or typological considerations of the the material culture (e.g., pottery). These approaches can have uncertainties of hundreds of years. Even in the fortunate circumstances of finding charcoal in direct association with the archaeomagnetic sample, the calibration of a given radiocarbon age into a calendar age is not always straightforward. Radiocarbon decays at a well determined rate, but the age depends not only on the parent/daughter ratio but on production rate of radiocarbon in the atmosphere and rates of sequestration into the deep ocean. While some radiocarbon ages have tightly constrained calibrations, others are very poorly constrained with uncertainties just from the calibration alone of some 400 years (as in the case of the Hallstatt plateau [9], which covers the period of interest of the current study, 800-400 BCE). Added to this problem is the fact that the charcoal could have come from a tree that was several hundred years old when it was cut down and burned (the "old wood effect").

The more conventional dating of archaeological contexts is based on typologies of material culture, especially ceramic and flint (e.g., [10]). This method also presents inherent difficulties, and usually provides an age range of a hundred years or even more. As an example of consequences of age uncertainties, there was relatively poor agreement between archaeomagnetic data from the Southern Levant and Northern Mesopotamia (mostly from Syria) until Shaar et al. [7] addressed this issue, and eliminated experimental design as the source of the problem. Instead, they found that the use of different methods to establish chronologies for the sampled artifacts—archaeological context and loose typologies for the Northern Levant and radiocarbon dated materials for the Southern Levant—was the culprit.

The current state of the archaeointensity database (included in the GEOMAGIA database of Brown et al. [11]) for the region contained within the bounds of latitude 27˚-40˚N and longitude 34˚-50˚E for the period from 2000 to 0 years BCE is shown in Fig 1. The majority of the data come from the Levantine Archaeomagnetic Curve (LAC) project, which was started through the efforts of Genevey et al. [16] in Syria and pursued by Ben-Yosef et al. [17] and other colleagues. The most recent version is that of Shaar et al. [18].

The increasingly detailed LAC is notable for its excellent age control and high quality paleomagnetic data. The LAC is based on data either from a so-called IZZI Thellier experiment as described in the following section here or from the Triaxe method [19] both of which have been thoroughly tested. Of particular interest is the period around 1000 BCE, the time of what has been termed the "Levantine Iron Age geomagnetic Anomaly" (LIAA, [18, 20, 21]) when there were periods of extremely high intensity values ("spikes", virtual axial dipole moments, VADMs, higher than 160 ZAm$^2$) and rapid changes in the field [21]. This phenomenon leads to large scatter in the data owing to discontinuous sampling in times of rapid change. While the existence of the 'spike' or 'spikes' is no longer hotly contested (e.g., [22]), there remains considerable uncertainty over how wide-spread the very high fields can be observed. For example, they are absent in data from Europe (see, e.g., [23]), an observation that led many to doubt the veracity of the spike itself (e.g., [24]). However, new data from Greece [25] do have relatively high VADMs ($\sim$ 140 ZAm$^2$) dated between 1070 and 1040 BCE that appear to be related to the LIAA to the east.

Because the source of the geomagnetic field is in the core, the spike cannot be a local, Levantine, phenomenon. While Shaar et al. [26] reported very high values from two samples from

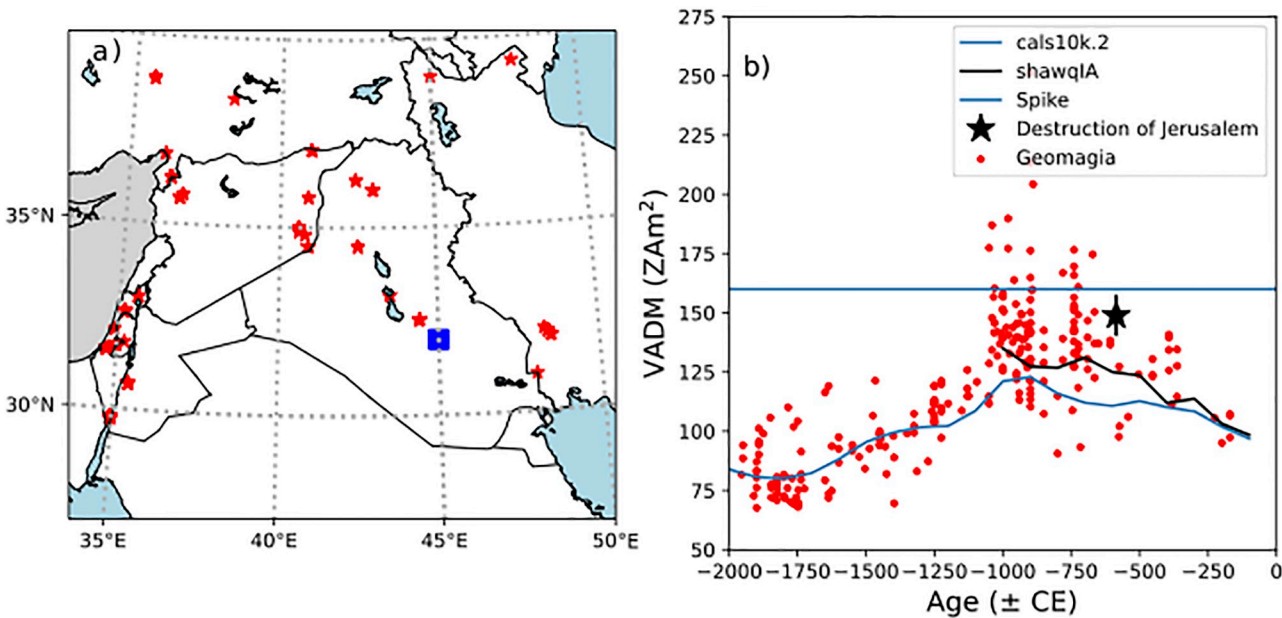

**Fig 1.** a) Red stars: Locations of data in the Geomagia database [11] with VADMs spanning -2000 to 0 CE. Blue square: Location of the Ishtar Gate (Babylon). b) Red dots: VADM values versus age. Blue line: model predictions from cals10k.2 model of [12]. Black line: model predictions from the shawqIA model of [13]. Dashed line is the threshold defined by [14] for a 'spike'. Black star are the results from the destruction of Jerusalem in 586 BCE of Vaknin et al. [15].

the Republic of Georgia at around 1000 BCE, the geographic extent of the LIAA continues to be poorly constrained owing to the limited high-quality data to the east. For example, there are only five data points in the GEOMAGIA database from Iraq (Fig 1a) for the entire period from 2000 to 0 BCE. The purpose of this paper is to expand the spatial extent of the archaeointensity database to the east, specifically, to Southern Mesopotamia and explore the use of fired bricks for archaeointensity research. At the same time, the purpose is to test whether the fast magnetic field variations can be used to better constrain the construction history of the Ishtar Gate using archeointensity techniques. In particular, we evaluate whether the three construction phases of the gate could have happened close in time (as would be suggested by similar archeointensity values) or not (as would be suggested by distinct archeointensity values) and also whether the Ishtar Gate was built near the time of the destruction of Jerusalem by King Nebuchadnezzar II in 586 BCE. Here we are fortunate that there are excellent archaeointensity results from Jerusalem's destruction layer itself by Vaknin et al. [15] (star in Fig 1b), which can be compared to the data from the Ishtar Gate obtained by the current study.

## Materials and methods

There are relatively few studies relying on fired bricks in the global paleomagnetic database known as MagIC (https://www2.earthref.org/MagIC). Mud bricks are the most common construction material in ancient Mesopotamia (e.g., [27] and references therein), and the use of fired mud bricks for construction commenced in this region at least during the Bronze Age if not before (ibid.). Moreover, from the middle of the third millennium BCE onward, we witness the appearance of fired mud bricks inscribed with names of particular kings of whom we often have historical information regarding the exact years of their reign. These bricks have the potential to contribute geomagnetic intensity data with excellent age constraints. In case of

glazed bricks, it might be possible to extract also the inclination; this has not been tested in the current study.

In order to test whether the Mesopotamian bricks can retain a reliable record of the ancient magnetic field, we obtained samples from a total of five bricks from the Ishtar Gate (Iron Age Babylon, see example in Fig 2). We sampled bricks from all three construction phases of the gate complex [28, 29], in order to potentially shed new light on the chronology of the gate's construction, in case reliable geomagnetic intensity are extracted.

The Ishtar Gate was constructed by order of King Nebuchadnezzar II (605 to 562 BCE), who claimed to have decorated the Ishtar Gate "with baked bricks (colored with) shining blue glaze that have (representations of) wild bulls (and) mušhuššu-dragon(s) fashioned upon them" [30]. The excavated remains of the gate complex reveal that he had the Ishtar Gate built several times during his reign. The various rebuilding projects are basically related to the new construction of the city fortifications and the remodeling of the adjacent palace area under Nebuchadnezzar II. In the process, the street level had to be significantly raised several times, which resulted in the gate having to be adapted as well, as the passage would otherwise have become too low, according to Nebuchadnezzar's II inscription (p 160-178 in [28], p 71-80 in

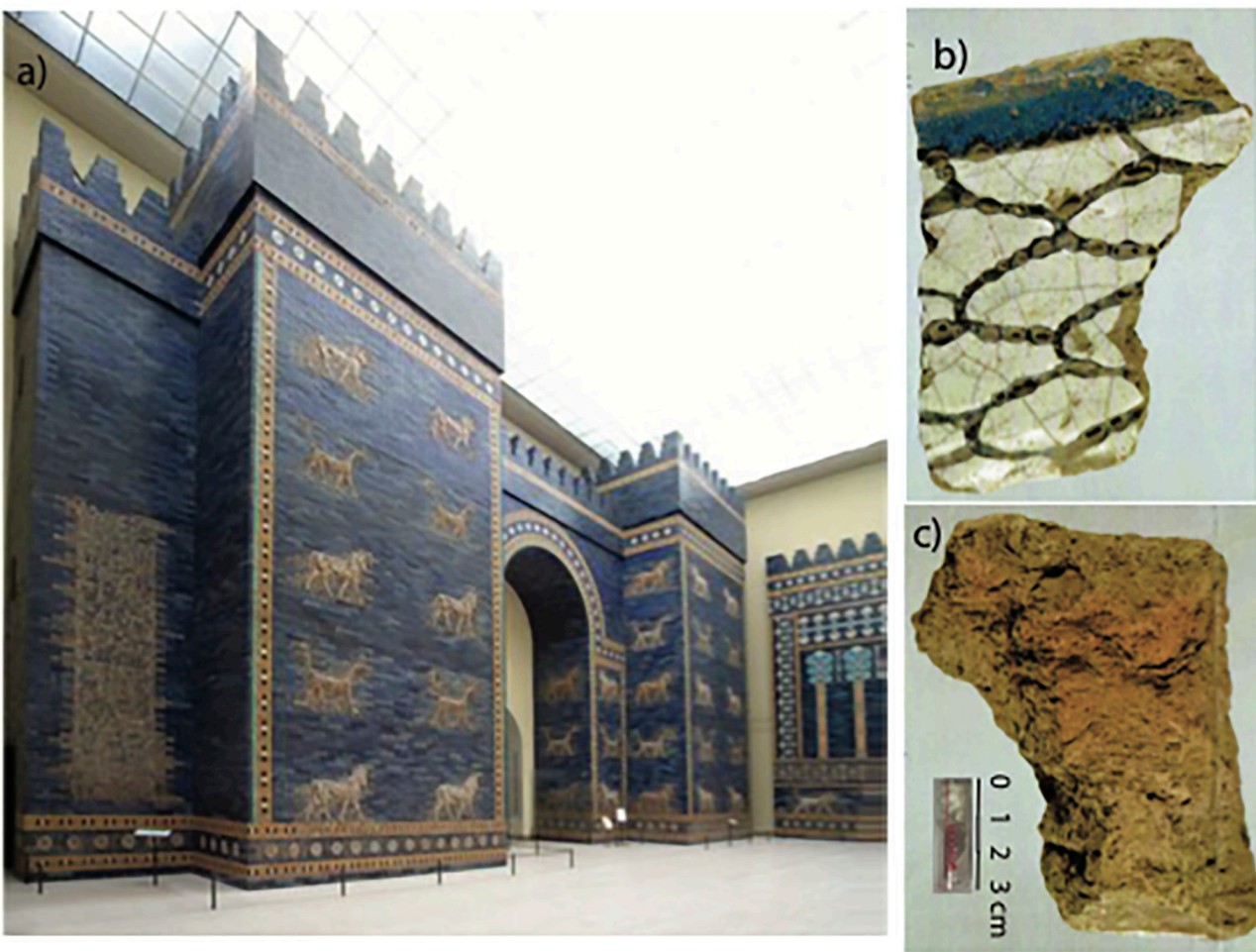

**Fig 2.** a) Ishtar Gate as reconstructed in the Pergamon Museum in Berlin, Germany. b) Brick (VA 17504) with blue glaze on the left hand side. Blue glaze was on the top of the brick. c) Back side of brick. Small fragments (0.1 gm) were taken from the brick and placed in specimen tubes like that shown in the inset. Photos, with permission, from: Staatliche Museen zu Berlin, Vorderasiatisches Museum / Olaf M. Teßmer.

**Table 1. Number of samples/specimens from bricks from the Ishtar Gate (courtesy of the Pergamon Museum) and their archaeological contexts with Phase I being the earliest.**

| Brick number | Context | Samples | Specimens |
|---|---|---|---|
| VA 17454 | Phase II | 3 | 4 |
| VA 17504 | Phase III | 5 | 6 |
| VA 17505 | Phase I | 5 | 8 |
| VA 17533 | Phase II | 1 | 9 |
| VA 17534 | Phase III | 1 | 5 |

[29, 30]). Due to their poor state of preservation, dating the individual construction phases of the Ishtar Gate is very difficult. The only anchors for the dating of the phases are the bricks inscribed with the name of Nebuchadnezzar II found in situ in Phase I and the finding of an archive above the adjoining street level 5, according to which the third subphase of the oldest phase (Phase I) could not have been erected before 592 BCE (p 74-75 in [29]). In addition, it is uncertain whether all other phases were constructed immediately one after the other, or whether there were any chronological gaps between them. It is even possible that the latest phase may not have been constructed under Nebuchadnezzar II, but later [31]. Parts of this phase of the gate were reconstructed in the Pergamon Museum in Berlin in 1930 [32].

We took samples of three brick fragments from the collection of Vorderasiatisches Museum Berlin. The fragments belong to the different types of building decoration of the Ishtar Gate, representing the three main construction phases of the gate. The oldest phase consisted of unglazed molded bricks and based on the adjacent street levels, this phase can probably be divided into up to four subphases (p 165-167 in [28]). The second main phase consisted of flat glazed bricks, and the third phase glazed molded bricks (Fig 2c and Table 1, [33]). The Babylonians glazed bricks in a skillful manner; they mastered producing brightly colored glazes in hues of white, black, green, yellow, orange, turquoise, and blue in large quantities [34]. The characteristic deep blue tint, obtained using cobalt oxide as a coloring agent, was added for the first time to glaze during the Neo-Babylonian Period [35]. None of the fragments used for this study were found in situ and assignment to phases was based on the type of brick decoration. We obtained tiny fragments (2-10 mm, "samples") from the back sides of the analyzed bricks. The samples were broken into 0.1 gm specimens and glued into specimen tubes (inset to Fig 2c) which were scribed with a fiducial line and a laboratory specimen identification name. Details of the bricks, the number of samples and specimens along with the construction phase (archaeological context) are given in Table 1.

### Archaeointensity analyses

Specimens from the Ishtar Gate (Table 1 and Fig 2) were subjected to the IZZI paleointensity experiment [5]. All the experiments were conducted in the Paleomagnetic Laboratory at Scripps Institution of Oceanography, University of California, San Diego. In the IZZI experiment, specimens are heated in a step-wise fashion, cooling either in an applied laboratory field (I steps) or in zero field (Z steps) at each temperature until at least 90% of the natural remanent magnetization (NRM) of each specimen was removed in the zero field steps. Zero-field cooling followed by in-field (ZI) or in-field cooling followed by zero field (IZ) alternate at every subsequent temperature step. In addition, an in-field step at a lower temperature was repeated after every IZ step to monitor for changes in the capacity of the specimens to acquire a partial thermal remanence (pTRM checks of [36]). The ratio of the natural remanence remaining compared to the pTRM gained over the experiment can be assumed to be quasi-linearly related to

the strength of the field in which the specimen acquired its NRM [37]. This ratio, when multiplied by the laboratory field $B_{lab}$, is taken as an estimate of the ancient field strength, $B_{anc}$. All successful specimens were also subjected to anisotropy of TRM experiments in which a total TRM was imparted while cooling in $B_{lab}$ in six directions. The average correction was negligible (0.996). Cooling rate corrections for a companion study on Mesopotamian bricks [38] were also negligible and were not carried out on the bricks from the Ishtar Gate.

There are many causes of failure of paleointensity experiments and the reliability of the results needs to be tested by quality criteria. Here we follow the Cromwell et al. [42] selection criteria (Table 2), called CCRIT by Tauxe et al. [43]. For a detailed explanation of what these criteria are, please see [44]. The criteria were designed to test the assumptions of the paleointensity experiment. Cromwell et al. [42] applied the CCRIT criteria to specimens taken from historical lava flow tops that cooled quickly in fields known from historical measurements and tabulated in the International Geomagnetic Reference Field models (e.g., [45]). The Cromwell et al. (2015) study recovered the field strength to within a few $\mu$T of the known field. CCRIT specifies threshold values for parameters at the specimen and at the site (e.g. cooling unit) levels. At the former, they are meant to test whether the demagnetization direction decays toward the origin using the deviation angle (DANG) and free-fitting maximum angle of deviation (MAD) parameters. DANG estimates the angle between the best fit line and the origin for the demagnetization direction. MAD measures the scatter in the NRM directions during the experiment. The ratio relating the remanence remaining against that acquired in the laboratory is estimated by the best fitting line through a selection of the data. For this study, we used the 'Auto Interpreter' function of the Thellier GUI program of [46], part of the PmagPy software package of [44] to find the portion of the data that passes the CCRIT criteria in an objective and reproducible way. Thellier_GUI finds all the ranges of temperature steps for a given specimen that satisfy the CCRIT criteria. The auto interpreter then estimates an average intensity for the collection of specimens (a site) with passing values that minimizes the uncertainty at the site level. The CCRIT threshold value for the standard deviation at the site level is 4 $\mu$T or 10% of the mean values.

The primary cause of failure in our archaeointensity experiments was because of curvature in the Arai plots (quantified with the $|\vec{k}|$ criterion of [6]). Phases with specimens showing a range of curvatures might contain useful information for constraining paleointensity estimates, particularly if there are many specimens at the site level. Here we apply the recently developed Bias-Corrected Estimation of Paleointensity (BiCEP) method of [41]. This method uses a Bayesian statistical approach, making the assumption that curved results ($|\vec{k}| > 0.164$) are linearly biased with respect to the true value as suggested by [47, 48]. Because there were more than five specimens from each phase, we subjected them to the BiCEP method (Figs 3c, 4c and 5c). Examples of curvature fits to the data from one specimen are shown in Fig 5a as thin green lines and the collection of estimates at the site level are shown in Fig 3c. The Bayesian credibility intervals give a range in estimates of 67.5-76.7 $\mu$T, in agreement with the CCRIT results but with a tighter credibility interval. These bounds are minimum and maximum

**Table 2. The CCRIT [42] selection criteria applied to the data from the IZZI experiment.** See [43] for expanded definitions. n: minimum number of consecutive demagnetization steps, DANG: deviation angle, MAD: maximum angle of deviation, $\beta$ = the maximum ratio of the standard error to the best fit slope, SCAT: a boolean value that indicates whether the data fall within $2\sigma_{threshold}$ of the best fit slope, FRAC: fractional remanence, $G_{max}$: maximum fractional remanence removed between consecutive temperature steps, $\vec{k}$: maximum curvature statistic, N: minimum number of specimens per site, B_%: maximum percentage deviation from the site average intensity, B_$\sigma$: maximum intensity ($\mu$T) deviation from the site average intensity.

| n | DANG | MAD | $\beta$ | SCAT | FRAC | $G_{max}$ | $|\vec{k}|$ | N | B_% | B$\sigma$ |
|---|---|---|---|---|---|---|---|---|---|---|
| 4 | $\leq 10°$ | $\leq 5°$ | 0.1 | TRUE | 0.78 | $\geq 0.6$ | 0.164 | 3 | 10 | 4 $\mu$T |

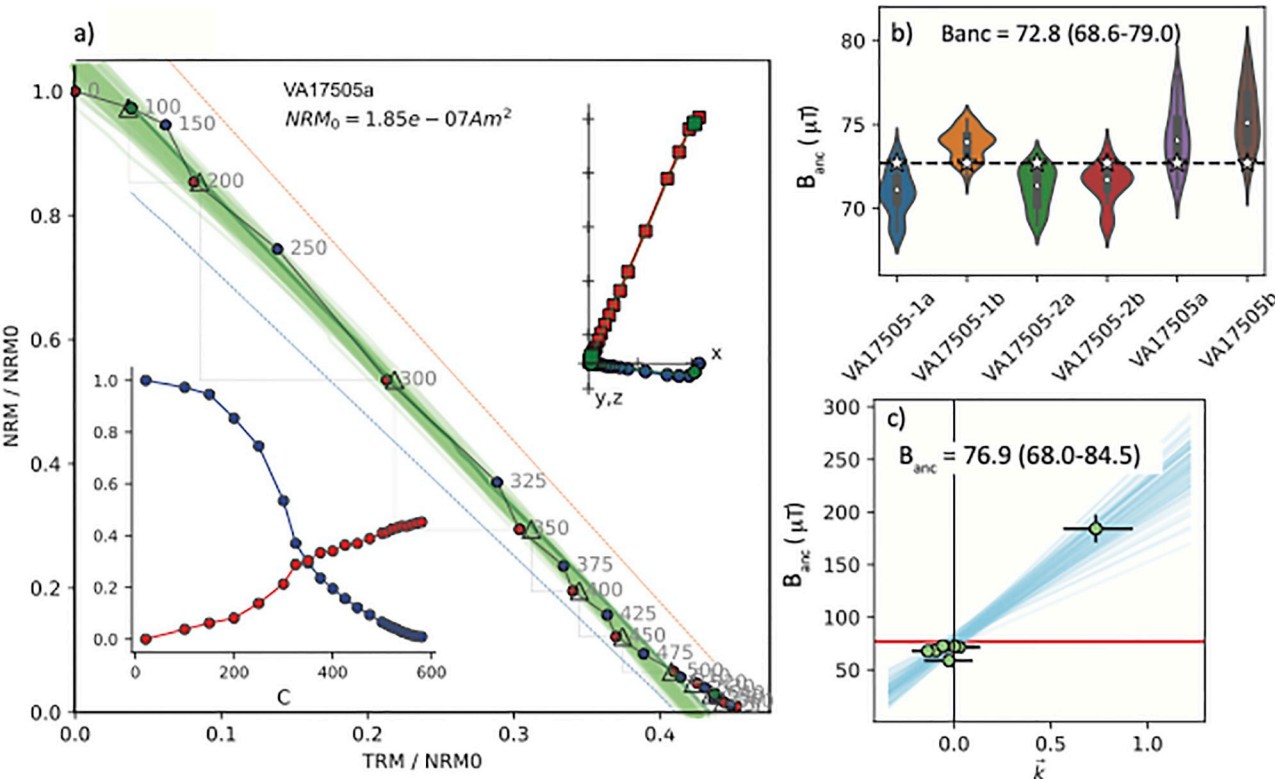

**Fig 3. Example of archaeointensity results for the Phase I brick (VA 17505, see Table 1).** a) Arai plot [39] from IZZI experiment for a representative specimen. Grey numbers are the temperature steps (in °C) with blue and red circles representing ZI and IZ steps respectively. Zijderveld and magnetization versus temperature (M/T) plots are shown as inserts to the upper right and lower left of each diagram respectively. The Zijderveld [40] plot from unoriented specimen with blue circles in the x,y plane and red squares in the x, z plane. b) Summary plot with estimated $B_{anc}$ for each specimen that passed the CCRIT criteria plotted as 'violins' which are the kernel density plots showing the distributions of the accepted results for each brick. The white stars are the $B_{anc}$ chosen by Thellier GUI autointerpreter as the specimen result that passes CCRIT and minimizes the standard deviation at the site (brick) level. The mean of all specimen interpretations selected by CCRIT (plotted as white stars) is 72.8 $\mu$T and the range of all estimates passing CCRIT is 68.6-79.0 $\mu$T. c) BiCEP [41] results. Blue lines are BiCEP estimates of $B_{anc}$ versus $\vec{k}'$ for Monte Carlo samples. Vertical and horizontal lines are uncertainties in $B_{anc}$ versus $\vec{k}'$, respectively. $B_{anc}$ for the site (Phase I) is 76.9 $\mu$T given as the minimum and maximum credible intervals ranging from 68.0-84.5 $\mu$T.

estimates which are analogous to 95% confidence bounds (so four times the width of our $1\sigma$ uncertainties with CCRIT).

## Results and discussion

We plot results from the bricks by construction phase in Figs 3–5. An example of an Arai plot [39] which passed the CCRIT criteria is shown in Fig 3a. Fig 3b shows the results of the six (out of eight) specimens that passed the specimen level CCRIT criteria, with total range of results that passed CCRIT at the specimen level. Taking the specimen estimates that minimize scatter at the phase level (as is the custom in the CCRIT approach), these yielded an average intensity value of 72.8 $\mu$T with a range in the 'extended error bars' of 68.6-79.0 $\mu$T. The standard deviation of the Phase I brick specimen estimates that minimizes scatter at the phase level passes the CCRIT criteria of 4 $\mu$T. Specimens from Phase II (Fig 4) also behaved quite well, passing the CCRIT criteria with an estimated intensity of 73.2 $\mu$T and range of 63.0-83.4 ($\mu$T). Those from Phase III (Fig 5) did not pass CCRIT because the standard deviation of the six (out of 11) specimens exceeded the CCRIT site level criterion of $\pm 4\mu$T.

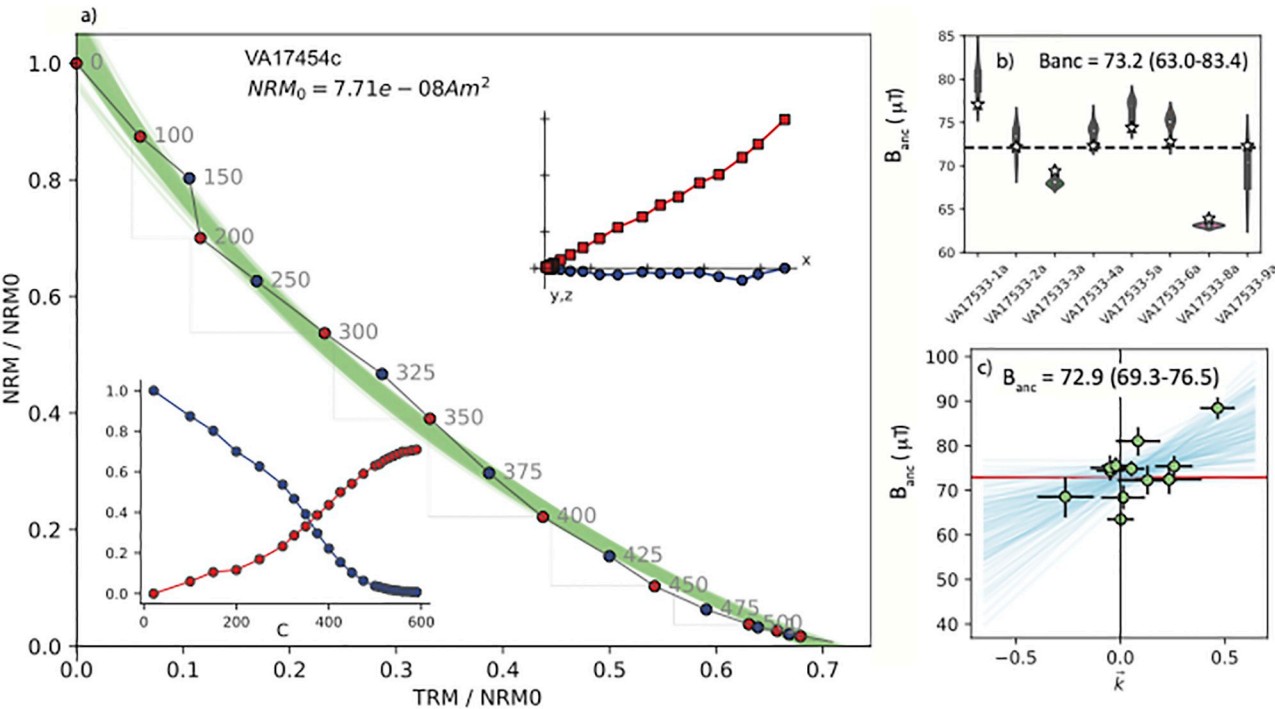

**Fig 4. Same as Fig 3 but for the Phase II bricks VA 17454 and VA 17533 (see Table 1).**

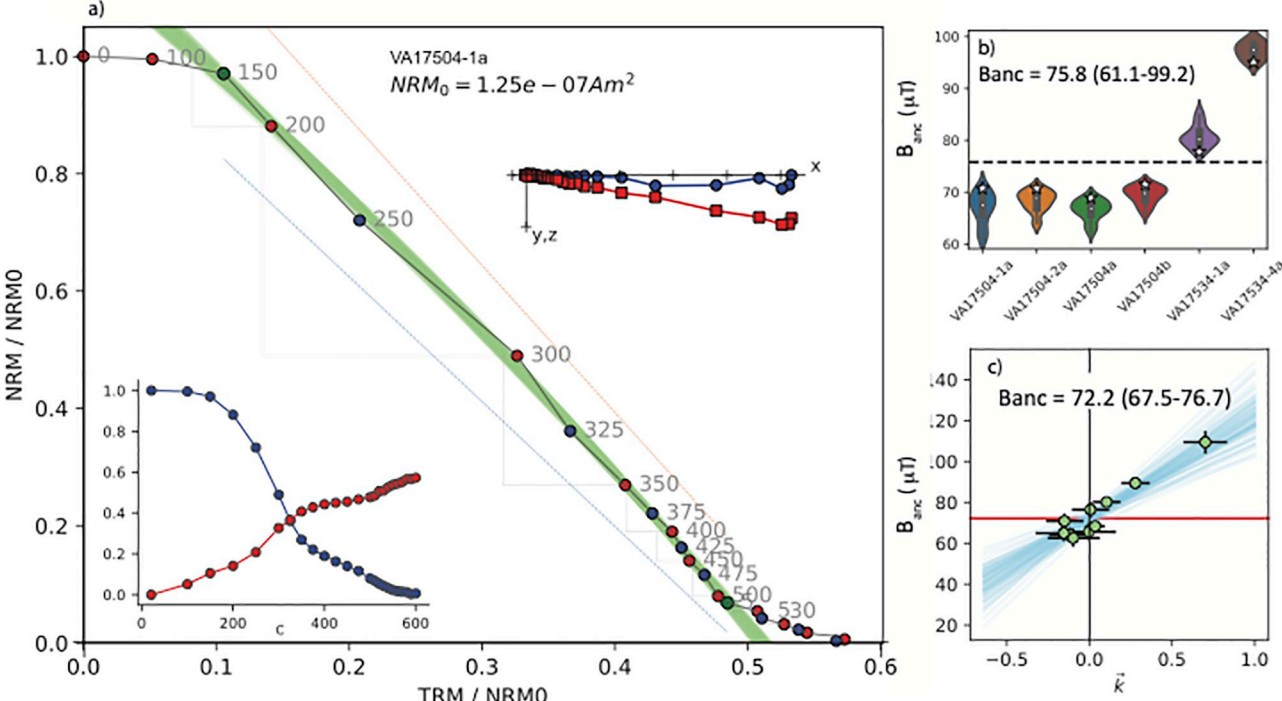

**Fig 5. Same as Fig 3 but for the Phase III bricks (VA 17504, shown in Fig 2b and 2c and VA 17534, and see Table 1).**

The results from all three phases are statistically indistinguishable and therefore we combined all of the specimens from the Ishtar Gate bricks into a single 'site' and treated them to the BiCEP analysis method (Fig 6a). The 95% credibility interval for the combined result is 67.8-76.3 $\mu$T. In order to compare results from geographically separated locations, it is customary to convert the values of the geomagnetic field (expressed in $\mu$T) to virtual axial dipole moments (VADMs) in ZAm$^2$. The data from the combined results is therefore 136±2.1 ZAm$^2$ (1$\sigma$ standard deviation). This estimate is much better constrained, based on the standard deviations, than that estimated from CCRIT (10.6 ZAm$^2$). Performing the auto interpreter for all the specimens taken together had 22 passing specimens (Fig 6b). These have a mean of 74 $\mu$T with a $\sigma$ of 5.6 or 7.5%, which passes the CCRIT site level criteria, but is less precise than the BiCEP result which used 30 of the specimens analyzed.

In summary, we obtained a high quality intensity data point for Southern Mesopotamia of 136±2.1 ZAm$^2$, with a narrow age range of 583±22 BCE. The age is based on the period of the reign of Nebuchadnezzar II, during which the order to build the gate was given. In addition, further examination of the magnetic results provide insights into the history of the construction of the gate complex. First, the statistical similarities of specimens from all three phases of the gate suggest that they were built with no significant chronological gaps between them, all of them during the period of Nebuchadnezzar II's reign, and most probably immediately one after the other. In other words, phases II and III are related to the original design of the gate and reflect the construction process rather than later additions, detached from the original construction of phase I.

Another observation is related to the exact date of the gate's construction within the period of Nebuchadnezzar II's reign. For this we use data from the Levant, most of which are from sites located less then 1000 km away (for example, Jerusalem is located $\sim$ 870 km west of Babylon). The Levantine archaeomagnetic curve (LAC) for the Bronze and Iron Ages is by now very well established, the culmination of decades of efforts by several teams (e.g., [16, 17, 22, 49, 50]). We plot the current version of the LAC and its uncertainty bounds in Fig 7a for the period from 2000 to 0 BCE. On Fig 7b, we plot the BiCEP results for the Ishtar intensities as a pair of red lines. The horizontal line is the reign of Nebuchadnezzar II and the vertical line spans 1$\sigma$ uncertainty (see Table 3) for the field strength obtained from the data in Fig 6a,

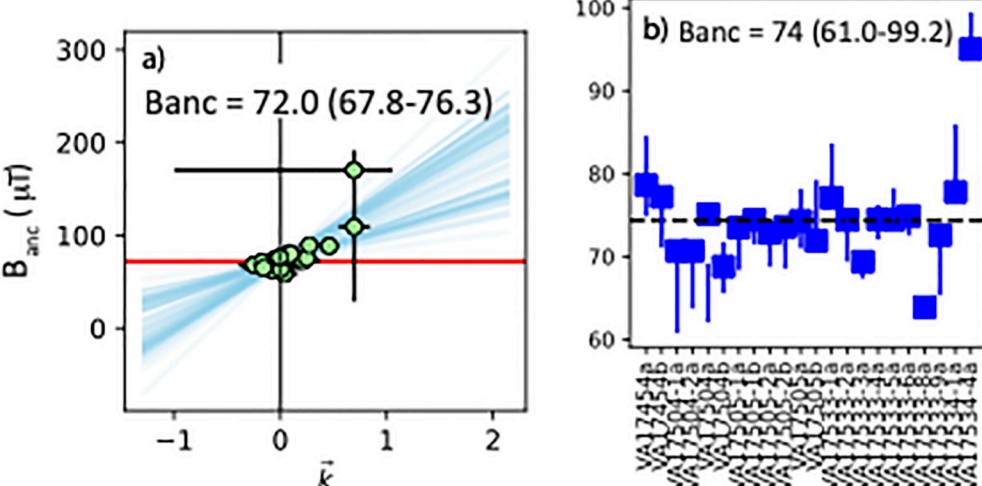

**Fig 6.** a) Same as Fig 2c, but for all specimens combined as a 'site'. b) Same as Fig 2b, but for all specimens that passed CCRIT.

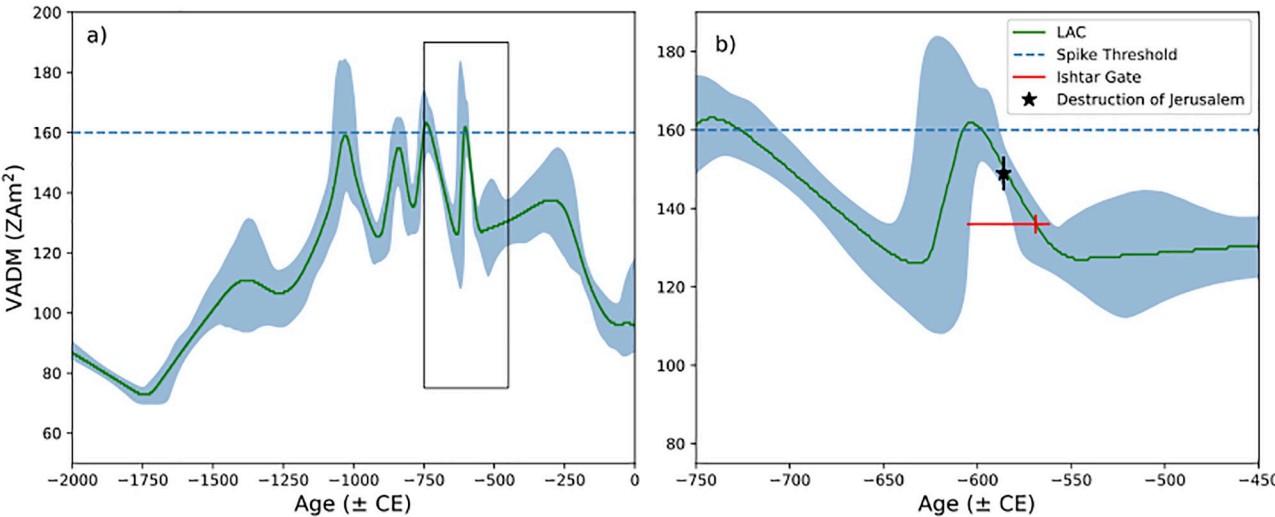

**Fig 7.** a) The Levantine archaeomagnetic curve (LAC) of Shaar et al. [22]. Box is bounds for b). b) LAC with results from Ishtar bricks as red lines. The horizontal red line is the duration of the reign of Nebukadnedzer and the vertical red line is the $1\sigma$ confidence bounds of the VADMs from BiCEP (see Table 3 on the point of maximum agreement with the LAC (569 BCE). The black star are the results from the Jerusalem destruction layer of Vaknin et al. [15] with $1\sigma$ confidence bounds.

converted to Virtual Axial Dipole Moment (VADM) in units of $ZAm^2$. The vertical line is placed at 569 BCE, which is where the mean crosses the LAC. This proposed date for the construction of the gate supports the suggestion that the gate complex was built after the successful Babylonian campaign to Judah and Jerusalem in 586 BCE [51]. However, as the recorded intensity for the time of the gate's construction ($136\pm2.1$ $ZAm^2$) is significantly different than the one recorded for the time of Jerusalem's destruction layer ($148.9\pm3.9$ $ZAm^2$, [15]) based on a Student's t-test p-value of $10^{-18}$, we should assume a certain chronological gap between the two events.

## Conclusion

In this study we reconstructed the ancient geomagnetic field intensity, as recorded in fired mud bricks used for the construction of the Ishtar Gate complex in Babylon. The experiments demonstrate that this type of material is an excellent recorder of the geomagnetic field, and

**Table 3. Phase: The construction phase for the Ishtar Gate bricks (see Table 1).** CCRIT ($\mu$T)/ ($ZAm^2$): Results of analysis using the CCRIT criteria [42] in microtesla ($B_{anc}$) and virtual axial dipole moment units (VADM). Uncertainty bounds for $B_{anc}$ are the range of values accepted by CCRIT (the 'extended error bars' of [46] and for the VADM values, uncertainties are $1\sigma$ calculated from the 'best' estimates from each specimen that produces the minimum scatter at the site level as is the practice in CCRIT. BiCEP ($\mu$T) / ($ZAm^2$): same as CCRIT but using the BiCEP analysis [41]. Uncertainty bounds for $\mu$T are Bayesian credibility intervals (analogous to the extended error bars from CCRIT). For the VADM values, we use the 95% credibility range divided by four, which is analogous to $1\sigma$ uncertainties used by CCRIT.

| Phase | CCRIT ($\mu$T) / ($ZAm^2$) | BiCEP ($\mu$T) / ($ZAm^2$) |
|---|---|---|
| Phase I | $72.8_{68.6}^{79.0}/138 \pm 0.47$ | $76.9_{68}^{84.7}/146 \pm 8.0$ |
| Phase II | $73.2_{63.0}^{83.4}/139 \pm 6.76$ | $72.9_{69.4}^{76.5}/138 \pm 3.4$ |
| Phase III | $75.8_{61.1}^{99.2}/143 \pm 18.7$ | $72.2_{67.5}^{76.7}/137 \pm 4.3$ |
| Combined | $74.0_{61}^{99}/140 \pm 10.6$ | $72.0_{67.8}^{76.3}/136 \pm 2.1$ |

that very small specimens ($< 3$ mm) are sufficient for extracting reliable geomagnetic information. This leads the way for future archaeomagnetic studies in southern Mesopotamia, a region in which a millennia-old tradition of construction with fired mud bricks exists. Moreover, in many cases these bricks bear inscriptions with names of kings whose ruling date is known to us. This provides the opportunity to reconstruct changes in the geomagnetic field in high age resolution, on a level which is usually not achievable by common archaeological dating methods, such as typology or radiocarbon. Using historically dated bricks for studying the ancient geomagnetic field is in particular potent for periods of plateaus in the radiocarbon calibration curve, such as the Hallstatt plateau that spans the period under consideration in the current study (800-400 BCE), in which radiocarbon dates can have an error range of $\pm$ 200 years [52].

The geomagnetic intensity value that we reconstructed from the Ishtar Gate (136±2.1 $ZAm^2$) also has a narrow age range based on historical information that ties the gate's construction to Nebuchadnezzar II, who reigned between 605 and 562 BCE. This makes the new data point an important anchor for models of the ancient magnetic field behavior in this specific region and beyond.

The magnetic information also helps elucidate the history of construction of the gate complex. While it was clear that phase I of the gate was indeed built by Nebuchadnezzar II (its bricks are inscribed with his name), the chronology of the other two phases was rather ambiguous, with suggestions ranging from no significant chronological gaps to the option that the later phase(s) were constructed after the reign of Nebuchadnezzar II. The statistical similarities of the magnetic results from all three phases strongly support the former option, i.e., that all phases were built during the reign of Nebuchadnezzar II and very close to each other, probably one immediately after the other.

Lastly, comparison of the results from the gate to data from the Levant suggests that the gate was built after the Babylonian conquest of Jerusalem in 586 BCE, although probably not immediately after, leaving the question whether or not it was erected to celebrate this victory open.

Providing observations on the geomagnetic field, archaeology, and history, this study demonstrates the multi-faceted contribution of archaeomagnetic studies, and the future potential of such studies to enhance both geophysical and archaeological investigations in Southern Mesopotamia, a region hitherto little explored through this avenue of research.

## Acknowledgments

Data will be made available on the MagIC database (earthref.org/MagIC/19876) upon publication of this article. The code used to analyze the data is available at https://github.com/PmagPy/PmagPy and https://github.com/bcych/BiCEP_GUI. This work was supported in part by BSF Grant 2018305 to LT and EB-Y. We are grateful for the assistance of Brendan Cych with BiCEP. We also appreciate the work of Christeanne Santos who made some of the measurements. Finally, we thank three anonymous reviewers and the editor for their useful comments that helped improving this manuscript.

No permits were required for the described study, which complied with all relevant regulations.

## Author Contributions

**Conceptualization:** Lisa Tauxe, E. Ben-Yosef.

**Data curation:** Anita Di Chiara, Lisa Tauxe, E. Ben-Yosef.

**Formal analysis:** Anita Di Chiara, Lisa Tauxe, Matthew D. Howland, E. Ben-Yosef.

**Funding acquisition:** Lisa Tauxe, E. Ben-Yosef.

**Investigation:** Anita Di Chiara, Lisa Tauxe, Matthew D. Howland, E. Ben-Yosef.

**Methodology:** Anita Di Chiara, Lisa Tauxe, E. Ben-Yosef.

**Project administration:** Lisa Tauxe, E. Ben-Yosef.

**Resources:** Lisa Tauxe, Helen Gries, Barbara Helwing, E. Ben-Yosef.

**Software:** Lisa Tauxe.

**Supervision:** Lisa Tauxe, E. Ben-Yosef.

**Validation:** Anita Di Chiara, Lisa Tauxe, Matthew D. Howland, E. Ben-Yosef.

**Visualization:** Anita Di Chiara, Lisa Tauxe, Helen Gries, Barbara Helwing, Matthew D. Howland, E. Ben-Yosef.

**Writing – original draft:** Anita Di Chiara, Lisa Tauxe, E. Ben-Yosef.

**Writing – review & editing:** Anita Di Chiara, Lisa Tauxe, Helen Gries, Barbara Helwing, Matthew D. Howland, E. Ben-Yosef.

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
