## [Decision Letter · Decision Letter 0]

19 Oct 2023

PONE-D-23-22213An archaeomagnetic study of the Ishtar Gate, BabylonPLOS ONE

Dear Dr. Tauxe,

Thank you for submitting your manuscript to PLOS ONE. After careful consideration, we feel that it has merit but does not fully meet PLOS ONE’s publication criteria as it currently stands. Therefore, we invite you to submit a revised version of the manuscript that addresses the points raised during the review process.

We noticed that an accept decision was sent to you in error on October 3. In order to allow you to address the reviewer comments and revise the manuscript before publication, we have rescinded the original decision and issued this decision as a minor revision instead. You will now be able to submit your revised manuscript, tracked changes and responses to the reviewers as normal, and the Academic Editor will be able to review the revised manuscript before it is sent to production.

We look forward to receiving your revised manuscript.

Kind regards,

Hanna Landenmark

Senior Editor, PLOS ONE

on behalf of 

Joe Uziel

Journal Requirements:

2. In your manuscript, please provide additional information regarding the specimens used in your study. Ensure that you have reported human remain specimen numbers and complete repository information, including museum name and geographic location. 

For more information on PLOS ONE's requirements for paleontology and archeology research, see https://journals.plos.org/plosone/s/submission-guidelines#loc-paleontology-and-archaeology-research.

3. Please expand the acronym “US” (as indicated in your financial disclosure) so that it states the name of your funders in full.

"Data will be made available on the MagIC database (earthref.org/MagIC/19876) upon

publication of this article. For the purposes of review, the data are available

at:https://earthref.org/MagIC/19876/8b970b5b-39bd-4e39-81c6-6e8047e6b20a. The

code used to analyze the data is available at https://github.com/PmagPy/PmagPy and

https://github.com/bcych/BiCEP GUI. This work was supported in part by BSF Grant

2018305 to LT and EBY. We are grateful for the assistance of Brendan Cych with

BiCEP. We also appreciate the work of Christeanne Santos who made some of the

measurements."

"This work was supported in part by US-Israelli Binational Science Foundation  Grant (bsf.org.il) 2018305 to LT and EBY. The funders had no role in study design, data collection and analysis, decision to publish, or preparation of the manuscript."

6. We note that Figure 1 in your submission contain [map/satellite] images which may be copyrighted. All PLOS content is published under the Creative Commons Attribution License (CC BY 4.0), which means that the manuscript, images, and Supporting Information files will be freely available online, and any third party is permitted to access, download, copy, distribute, and use these materials in any way, even commercially, with proper attribution. For these reasons, we cannot publish previously copyrighted maps or satellite images created using proprietary data, such as Google software (Google Maps, Street View, and Earth). For more information, see our copyright guidelines: http://journals.plos.org/plosone/s/licenses-and-copyright.

Additional Editor Comments:

This is an important contribution by the authors who have been involved in building the Levantine archaeomagnetic curve, which is becoming a necessary, precise tool for dating in the region. The current paper expands on previous work by the authors (leading scholars in the field of paleomagnetism) both geographically and methodologically in terms of the materials used. This will clearly be an important paper and worthy of publication in PlosOne, subsequent to minor revisions offered by 3 reviewers, as well as 2 additional comments I offer for the authors' consideration:

Lines 210-213: Although I agree with the caution on determining the construction of the gate as a result of the victorious siege over Jerusalem, this is almost mentioned in passing. I believe this discussion could be expanded, in considering the dating of the gate and in relation to other successful military accomplishments attributed to Nebuchadnezzar (the Battle of Carchemish, the conquest of Ashkelon and Ekron, the battle over Tyre).

Lines 224-225: Significant advances have been made in the radiocarbon calibration, which when used together with micro-archaeological sampling and stratigraphy, allows for very narrow dating windows.

Reviewers' comments:

Reviewer's Responses to Questions

**Comments to the Author**

1. Is the manuscript technically sound, and do the data support the conclusions?

Reviewer #1: Yes

Reviewer #2: Yes

Reviewer #3: Yes

2. Has the statistical analysis been performed appropriately and rigorously? 

Reviewer #1: I Don't Know

Reviewer #2: I Don't Know

Reviewer #3: Yes

3. Have the authors made all data underlying the findings in their manuscript fully available?

Reviewer #1: Yes

Reviewer #2: Yes

Reviewer #3: Yes

4. Is the manuscript presented in an intelligible fashion and written in standard English?

Reviewer #1: Yes

Reviewer #2: Yes

Reviewer #3: Yes

5. Review Comments to the Author

Reviewer #1: The following notes pertain to the historical (Assyriological) aspects of the paper:

The mention of Nebuchadnezzar’s instructions to build the gate should be properly referenced. Footnote 28 directs the reader to Kaniuth 2018, but it lacks specific page references. Kaniuth (2018: 350) himself directs the reader to two (entire) books (Berger 1973 and Da Riva 2008), with no further specification (neither page nor text numbers are given). It would be more useful to refer the reader to Da Riva 2008: 118 § 2.1b, where the specific inscriptions are listed with references to Berger.

Regarding the statement: “it was even considered that the latest phase was constructed under Nebuchadnezzar II” (p. 10 (4/11), ll. 96–97); I suspect that what was meant is actually: "construction POST Nebuchadnezzar II," which would make more sense in this context; cf., l. 235, p. 14 (8/11), where the authors indeed use “post Nbk.”

Concerning the three phases of the gate, since this paper is of interest to scholars from various backgrounds (the present reviewer is an Assyriologist, for example), it may be helpful to include a brief explanation of the nature of the three archaeological phases. It is important to explain to the reader the basis for distinguishing the different phases, such as different construction techniques, materials used, or the stylistic aspects of the glaze.

Understandably, the identification of three phases raises questions about possible chronological gaps between them. It may also be worth mentioning, however, that (to the best of my recollection, without further investigation) no other king is known to have claimed responsibility for building the gate.

Regarding p. 13 (7/11), ll. 205–213: I do not have access to Fitzgerald/Knott 2019 (FN 43), so I could not follow the referenced claim (here too, parenthetically, no page numbers are given, just the book). Regardless, I am not familiar with the suggestion that the Ištar Gate was a “triumphal arch” erected to commemorate the 586 Jerusalem campaign. This notion appears peculiar and problematic (in its Babylonian context), seemingly resonating with Roman practices, Bibliocentrism, or both(?). Even if this argument is presented in/by Fitzgerald/Knott 2019, it would be advisable to provide further elaboration and, I would add, exercise caution in presenting such an assertion.

Reviewer #2: This is important original pioneering research using the magnetic intensity for dating archaeological baked brick from the Ishtar Gate in ancient Babylon. The study is provisional with a lack of localized comparative results therefore instead relying essentially on the more established Levatine Archaeomagnetic Curve. As described in the paper, many baked bricks from the area have inscriptions allowing the historic date, and it should therefore be rather easy to continue with more analysis of bricks producing a local archaeomagnetic curve.

There are some historical and archaeological problem with the paper. The three phases of the gate discussed in the paper are the three types of wall decoration as presented in the museum and are not all the archaeologically attested levels of rebuilding the gate during the reign of Nebuchadnezzar II (605-562). The archaeological excavations as documented in R. Koldewey, 1918, Das Ischtar-Tor in Babylon, and summarized with updated material from cuneiform inscriptions by O. Pedersén in references [26] and [27] show a gate connected with the city wall rebuilt on much higher terrain several times during the 43-year reign of Nebuchadnezzar. He probably started the first version of the gate quite early in his reign, the first main rebuilding after raising the processional street through the gate several meters have a date 592 BC according to cuneiform texts. Later in his reign the street and gate were raised and rebuilt several times. The samples tested are probably all from rebuildings of the gate late in the reign of Nebuchadnezzar even if it is not said in the paper from where the used Phase I brick was taken.

The resultant date 569 BC is reasonable, but provisional until a local archaeomagnetic curve has been established. It disproves the suggestion in reference [28] that the latest phase was constructed after Nebuchadnezzar. Here the manuscript has a typing mistake: it says *under* and not the correct *after* Nebuchadnezzar.

The correct understanding of the subsequent rebuildings of the gate during the long reign of Nebuchadnezzar makes the discussions if the gate was built before or after the destruction of Jerusalem irrelevant as both are true. It would be better to take this a bit speculative part away before publication.

Would it be possible to name the lab where the test have been made?

Reviewer #3: This article meets the quality standards for PLOS ONE. It addresses an interesting issue as the investigation of intensity changes for a region relatively unexplored so far, such as Southern Mesopotamia. Furthermore, the paper explores the chronological and historical implications of the results obtained. The new archaeomagnetic data have been obtained from very well-dated bricks following standard procedures and are of good quality. These new data provide unique information about past values in geomagnetic field intensity for this region and time interval. The subsequent analysis and conclusions drawn from the data are consistent and well-presented. The paper is well-written, well-organized and the quality of the figures is high. I therefore recommend publication after minor changes. I have, however, some concerns that should be addressed before the final publication:

• Lines 47-57. I would like to read a more detailed description of the quality of the data available for the region.

• Line 56: This statement is not accurate. It is true that in western Europe, this peak does not appear to be present but a recent study by Rivero-Montero et al. (2021) demonstrated a high-intensity peak is Greece between 1070 and 1040 BCE, suggesting its association to the LIAA. Please, include this information in the new version.

• Figure 2: the explanation of Figure 2 is missing (lines 105-106)

• Lines 106-119. What about TRM anisotropy and cooling rate corrections? Have these effects been measured? Nothing is said about these important issues. I know that the TRM anisotropy´s effect on bricks is generally low, but some details about this should be included. On the other hand, it is not explained if the cooling rate has been measured or considered. If this was not the case, do you consider that your mean intensity should be affected? How would your results and derived dating be modified by, for example, a mean value of 5% for the cooling rate effect upon paleointensity estimates?

• Do you have an explanation for the higher dispersion observed in Figure 4 results? It seems that some of the results might be affected by some effects such as MD behaviour since the trend seems not to be linear. It would be nice to show additional NRM-TRM plots.

• Lines 194-213: How reliable is the proposed date? If we take into account both the errors in the PSV reference curve used and the mean intensity of Ishtar Gate, an age interval is obtained instead of an exact date. An accurate archaeomagnetic dating anañysis should be included here, considering the uncertainties of both the curve and the data. Please, improve this section.

• In general, the quality of the figures in the pdf file is not high, but I assume that they should be ok when submitted as individual files.

• I also recommend including references from other international groups devoted to archaeomagnetic research to enrich the context. While the provided references are relevant, incorporating these citations would strength the paper´s global perspective.

I hope it helps.

Regards

6. PLOS authors have the option to publish the peer review history of their article (what does this mean?). If published, this will include your full peer review and any attached files.

Reviewer #1: No

Reviewer #2: No

Reviewer #3: No

---

## [Author Response · Author response to Decision Letter 0]

9 Nov 2023

November 9, 2023

Dear PLoS ONE editor,

We were happy to follow up on your and the reviewers’ comments and we hereby submit a revised manuscript. Below, please find our specific responses to the comments. 

We hope the paper will be accepted for publication and thank you very much for your efforts in the publication process,

Lisa Tauxe, on behalf of all co-authors

Journal Requirements:

In your manuscript, please provide additional information regarding the specimens used in your study. Ensure that you have reported human remain specimen numbers and complete repository information, including museum name and geographic location. 

Specific responses to October 27 letter from PLOS ONE: 

PONE-D-23-22213R1

An archaeomagnetic study of the Ishtar Gate, Babylon

Prof. Lisa Tauxe

We note that your manuscript is not formatted using one of PLOS ONE’s accepted file types. Please reattach your manuscript as one of the following file types: .doc, .docx, .rtf, or .tex (accompanied by a .pdf).

If your submission was prepared in LaTex, please submit your manuscript file in PDF format and attach your .tex file as “other.

I have now uploaded the .tex formatted file as requested. 

2. We note that Figure 1 in your submission contain [map/satellite] images which may be copyrighted. All PLOS content is published under the Creative Commons Attribution License (CC BY 4.0), which means that the manuscript, images, and Supporting Information files will be freely available online, and any third party is permitted to access, download, copy, distribute, and use these materials in any way, even commercially, with proper attribution. For these reasons, we cannot publish previously copyrighted maps or satellite images created using proprietary data, such as Google software (Google Maps, Street View, and Earth). For more information, see our copyright guidelines: http://journals.plos.org/plosone/s/licenses-and-copyright.

We've returned your manuscript to your account. Please resolve these issues and resubmit your manuscript within 21 days. If you need more time, please email the journal office at plosone@plos.org. We are happy to grant extensions of up to one month past this due date. If we do not hear from you within 21 days, we will withdraw your manuscript.

Please log on to PLOS Editorial Manager at https://www.editorialmanager.com/pone/ to access your manuscript. You will find your manuscript in the 'Submissions Sent Back to Author' link under the New Submissions menu. Be sure to remove your previous manuscript file if you are uploading a new file in response to these requests. After you've made the changes requested above, please be sure to view and approve the revised PDF after rebuilding the PDF to complete the resubmission process.

We are requesting these changes to comply with the PLOS ONE submission guidelines (https://journals.plos.org/plosone/s/submission-guidelines). Please note that we won't send your manuscript for review until you have resolved the above requests. 

Thank you for submitting your work to PLOS ONE and supporting our mission of Open Science.

Figure 1 was created with open source Python software by the authors of this paper and is not subject to copyright. 

Kind regards,

Adrian Cyrus Luczon

PLOS ONE

No permits were required; the relevant statement was added at the end of the manuscript.

Additional Editor Comments:

This is an important contribution by the authors who have been involved in building the Levantine archaeomagnetic curve, which is becoming a necessary, precise tool for dating in the region. The current paper expands on previous work by the authors (leading scholars in the field of paleomagnetism) both geographically and methodologically in terms of the materials used. This will clearly be an important paper and worthy of publication in PlosOne, subsequent to minor revisions offered by 3 reviewers, as well as 2 additional comments I offer for the authors' consideration:

Lines 210-213: Although I agree with the caution on determining the construction of the gate as a result of the victorious siege over Jerusalem, this is almost mentioned in passing. I believe this discussion could be expanded, in considering the dating of the gate and in relation to other successful military accomplishments attributed to Nebuchadnezzar (the Battle of Carchemish, the conquest of Ashkelon and Ekron, the battle over Tyre).

Thanks. This section has been revised and now includes more details (also see below). 

Lines 224-225: Significant advances have been made in the radiocarbon calibration, which when used together with micro-archaeological sampling and stratigraphy, allows for very narrow dating windows.

Yet, we still have several periods of plateaus in the calibration curve – including during the 6th c. BCE. We’ve added a note regarding this. 

Reviewer #1

The following notes pertain to the historical (Assyriological) aspects of the paper:

The mention of Nebuchadnezzar’s instructions to build the gate should be properly referenced. Footnote 28 directs the reader to Kaniuth 2018, but it lacks specific page references. Kaniuth (2018: 350) himself directs the reader to two (entire) books (Berger 1973 and Da Riva 2008), with no further specification (neither page nor text numbers are given). It would be more useful to refer the reader to Da Riva 2008: 118 § 2.1b, where the specific inscriptions are listed with references to Berger.

We have now included a reference directly to the inscription. However, not to these two publications, since these are outdated in our opinion, but to the current online reference work.

Regarding the statement: “it was even considered that the latest phase was constructed under Nebuchadnezzar II” (p. 10 (4/11), ll. 96–97); I suspect that what was meant is actually: "construction POST Nebuchadnezzar II," which would make more sense in this context; cf., l. 235, p. 14 (8/11), where the authors indeed use “post Nbk.”

Thanks. The entire section has been revised. 

Concerning the three phases of the gate, since this paper is of interest to scholars from various backgrounds (the present reviewer is an Assyriologist, for example), it may be helpful to include a brief explanation of the nature of the three archaeological phases. It is important to explain to the reader the basis for distinguishing the different phases, such as different construction techniques, materials used, or the stylistic aspects of the glaze.

We've added two more sentences and two more references so there's a little more context to the bricks now.

Understandably, the identification of three phases raises questions about possible chronological gaps between them. It may also be worth mentioning, however, that (to the best of my recollection, without further investigation) no other king is known to have claimed responsibility for building the gate.

Regarding p. 13 (7/11), ll. 205–213: I do not have access to Fitzgerald/Knott 2019 (FN 43), so I could not follow the referenced claim (here too, parenthetically, no page numbers are given, just the book). Regardless, I am not familiar with the suggestion that the Ištar Gate was a “triumphal arch” erected to commemorate the 586 Jerusalem campaign. This notion appears peculiar and problematic (in its Babylonian context), seemingly resonating with Roman practices, Bibliocentrism, or both(?). Even if this argument is presented in/by Fitzgerald/Knott 2019, it would be advisable to provide further elaboration and, I would add, exercise caution in presenting such an assertion.

We agree, and toned down the possible connection to the conquest of Jerusalem. We also changed the reference in regard to this issue. 

Reviewer #2: 

This is important original pioneering research using the magnetic intensity for dating archaeological baked brick from the Ishtar Gate in ancient Babylon. The study is provisional with a lack of localized comparative results therefore instead relying essentially on the more established Levatine Archaeomagnetic Curve. As described in the paper, many baked bricks from the area have inscriptions allowing the historic date, and it should therefore be rather easy to continue with more analysis of bricks producing a local archaeomagnetic curve.

There are some historical and archaeological problem with the paper. The three phases of the gate discussed in the paper are the three types of wall decoration as presented in the museum and are not all the archaeologically attested levels of rebuilding the gate during the reign of Nebuchadnezzar II (605-562). The archaeological excavations as documented in R. Koldewey, 1918, Das Ischtar-Tor in Babylon, and summarized with updated material from cuneiform inscriptions by O. Pedersén in references [26] and [27] show a gate connected with the city wall rebuilt on much higher terrain several times during the 43-year reign of Nebuchadnezzar. He probably started the first version of the gate quite early in his reign, the first main rebuilding after raising the processional street through the gate several meters have a date 592 BC according to cuneiform texts. Later in his reign the street and gate were raised and rebuilt several times. The samples tested are probably all from rebuildings of the gate late in the reign of Nebuchadnezzar even if it is not said in the paper from where the used Phase I brick was taken.

We have rewritten the corresponding passage and hope it is more understandable. 

The resultant date 569 BC is reasonable, but provisional until a local archaeomagnetic curve has been established. It disproves the suggestion in reference [28] that the latest phase was constructed after Nebuchadnezzar. Here the manuscript has a typing mistake: it says *under* and not the correct *after* Nebuchadnezzar.

Fixed.

The correct understanding of the subsequent rebuildings of the gate during the long reign of Nebuchadnezzar makes the discussions if the gate was built before or after the destruction of Jerusalem irrelevant as both are true. It would be better to take this a bit speculative part away before publication.

We have toned down this possible connection. 

Would it be possible to name the lab where the test have been made?

Added.

Reviewer #3: 

This article meets the quality standards for PLOS ONE. It addresses an interesting issue as the investigation of intensity changes for a region relatively unexplored so far, such as Southern Mesopotamia. Furthermore, the paper explores the chronological and historical implications of the results obtained. The new archaeomagnetic data have been obtained from very well-dated bricks following standard procedures and are of good quality. These new data provide unique information about past values in geomagnetic field intensity for this region and time interval. The subsequent analysis and conclusions drawn from the data are consistent and well-presented. The paper is well-written, well-organized and the quality of the figures is high. I therefore recommend publication after minor changes. I have, however, some concerns that should be addressed before the final publication:

• Lines 47-57. I would like to read a more detailed description of the quality of the data available for the region.

We added a sentence about this citing the Shaar and Gallet papers regarding methods. 

• Line 56: This statement is not accurate. It is true that in western Europe, this peak does not appear to be present but a recent study by Rivero-Montero et al. (2021) demonstrated a high-intensity peak is Greece between 1070 and 1040 BCE, suggesting its association to the LIAA. Please, include this information in the new version.

We added this now. thanks to the reviewer for alerting us to this publication. 

• Figure 2: the explanation of Figure 2 is missing (lines 105-106)

The caption is in the manuscript. We are not sure what is the source of the confusion.

• Lines 106-119. What about TRM anisotropy and cooling rate corrections? Have these effects been measured? Nothing is said about these important issues. I know that the TRM anisotropy´s effect on bricks is generally low, but some details about this should be included. On the other hand, it is not explained if the cooling rate has been measured or considered. If this was not the case, do you consider that your mean intensity should be affected? How would your results and derived dating be modified by, for example, a mean value of 5% for the cooling rate effect upon paleointensity estimates?

We added an explanation about this - that ATRM was done (and it was trivial) and that cooling rate was not done as the companion paper on a larger set of bricks by Howland et al. (in press) shows that was also negligible in all of those Mesopotamian bricks. 

• Do you have an explanation for the higher dispersion observed in Figure 4 results? It seems that some of the results might be affected by some effects such as MD behaviour since the trend seems not to be linear. It would be nice to show additional NRM-TRM plots.

As all data are in the MaGIC Database, these (and other) plots are readily available.

• Lines 194-213: How reliable is the proposed date? If we take into account both the errors in the PSV reference curve used and the mean intensity of Ishtar Gate, an age interval is obtained instead of an exact date. An accurate archaeomagnetic dating anañysis should be included here, considering the uncertainties of both the curve and the data. Please, improve this section.

The date and its errors were provided.

• In general, the quality of the figures in the pdf file is not high, but I assume that they should be ok when submitted as individual files.

We’ve submitted high resolution files, thanks.

• I also recommend including references from other international groups devoted to archaeomagnetic research to enrich the context. While the provided references are relevant, incorporating these citations would strength the paper´s global perspective.

We have in the manuscript basic references that include all the other relevant references for the archaeomag of this region. 

I hope it helps.

Thank you

---

## [Editor Report · Decision Letter 1]

7 Dec 2023

An archaeomagnetic study of the Ishtar Gate, Babylon

PONE-D-23-22213R1

Dear Dr. Tauxe,

We’re pleased to inform you that your manuscript has been judged scientifically suitable for publication and will be formally accepted for publication once it meets all outstanding technical requirements.

Kind regards,

Joe Uziel

Academic Editor

PLOS ONE
---

## [Editor Report · Acceptance letter]

27 Dec 2023

PONE-D-23-22213R1 

PLOS ONE

Dear Dr. Tauxe, 

I'm pleased to inform you that your manuscript has been deemed suitable for publication in PLOS ONE. Congratulations! Your manuscript is now being handed over to our production team.

Kind regards, 

on behalf of

Dr. Joe Uziel 

Academic Editor

PLOS ONE